# Ultra-Low Reflectivity in Visible Band of Vanadium Alumina Nanocomposites

Qiujin Wang [1,2], Guozhong Zhao [1] and Hai Wang [1,*]

1    Department of Physics, Capital Normal University, Beijing 100048, China
2    School of Physics and Astronomy, Yunnan University, Kunming 650091, China
*    Correspondence: wanghai@cnu.edu.cn

**Abstract:** The high-temperature annealing effect of a $V_2O_5/Al_2O_3$ bilayer on the reflectivity of visible band was studied in the present work. It was found that the $V_2O_5$ (200 nm)/$Al_2O_3$ (30 nm) sample (A-0) has a nano-granular surface morphology without a definite crystalline phase. The reflectance of A-0 overreaches 60% as the wavelength is larger than 650 nm, and its minimum reflectance (20%) occurs at the wavelength of ~500 nm. After in situ annealing treatment at 750 °C for 30 min, a triclinic $AlVO_4$ phase appears while the nano-granular surface morphology remains (sample A-30). The reflectivity of A-30 is well suppressed in comparison with that of A-0 especially in the long wavelength range. Its minimum reflectivity approaches 5% at ~500 nm wavelength. It is speculated that the appearance of a new $AlVO_4$ nanophase is the main reason for the decline of reflectivity. Extending the annealing time to 60 min, the reflectivity spectrum, as well as the surface morphology, are almost the same. These results can be helpful for applications of vanadium alumina nanocomposites in high-temperature environments.

**Keywords:** post-annealing; interfacial mutual-diffusion; magnetron sputtering; reflectivity; vanadium alumina nanocomposites; visible band

## 1. Introduction

Vanadium pentoxide ($V_2O_5$) has a layered crystalline structure with a weak interlayer van der Waals interaction. Micro- and nano-structured $V_2O_5$ have attracted a great deal of attention due to their promise in different applications such as lithium batteries [1,2], electrochromic devices [3], optoelectronic switches [4,5], catalyst material [6,7], terahertz luminescence [8], as well as the antireflection coating on solar cells [9]. In recent decades, much effort has been focused on the synthesis and the structural, optical, and electrical properties of $V_2O_5$ thin films.

$V_2O_5$ suffers from poor conductivity and stability for many applications and hence several studies investigated supported $V_2O_5$ to overcome these shortcomings [10]. The improvements in $V_2O_5$ performance as regards mechanical and thermal stability are also observed through supports such as $Al_2O_3$, $SiO_2$, $TiO_2$ [11]. The process of phase transition, the oxidation state change of $V_2O_5/Al_2O_3$ caused by temperature [12], as well as the surface quality of $Al_2O_3$ substrate are crucial for the structural dynamics in $V_2O_5/Al_2O_3$ nanocomposites [13]. The optimized deposition temperature of $V_2O_5$ thin films is higher than 300 °C [4,14,15], and the practical application of $V_2O_5/Al_2O_3$ nanocomposites or aluminium-based is always under relative high temperature environments (e.g., 500, 550, 800 °C, etc.) [12,16,17]. Such being the case, vanadium alumina nanocomposites emerge inevitably due to the interfacial mutual-diffusion between $V_2O_5$ and $Al_2O_3$ especially in nano-thin films. Synthesis and characterization of new materials is a very important topic for several applications [18–21]. In addition, it is known that the optical properties of the vanadium alumina nanocomposites are greatly affected by the crystalline structure and the deposition temperature [14,22]. The high deposition temperature (500 °C) prompts

the increase of the refractive index (n) of the sample [22]. However, there are few relevant studies on high-temperature influences on the optical properties of ultrathin vanadium-alumina nanocomposites films. In particular, there is a lack of research on the influence of the interfacial mutual-diffusion, as well as the accompanying aluminum-vanadium oxides on the reflection properties of vanadium alumina nanocomposites.

In the present work, we deposited the $V_2O_5$ thin films on $Al_2O_3$ at high temperature following a short-period in situ annealing. It was observed that triclinic $AlVO_4$ in vanadium alumina nanocomposites appears after a short period of annealing. The reflectivity of vanadium alumina nanocomposites was successfully suppressed in a visible band and the minimum reflectivity approached to ~5% at 500 nm wavelength. It is considered that the appearance of metallic nanograins of $AlVO_4$ in vanadium alumina nanocomposites is a non-negligible crystalline change.

## 2. Experiments

Vanadium alumina nanocomposites were fabricated by a two-step magnetron sputtering method. The base pressure of vacuum chamber and the working pressure of argon gas (99.999%) were $1.5 \times 10^{-5}$ Pa and 0.3 Pa, respectively. Firstly, a 30-nm-thick $Al_2O_3$ layer was deposited on the cleaned silicon (100) ($\Phi 50.8 \pm 0.2$ mm) wafers by radio frequency magnetron sputtering (200 W) from an $Al_2O_3$ target at room temperature. Then, the deposition temperature was elevated to 750 °C, which held steady during the subsequent deposition process. Secondly, a 200-nm-thick $V_2O_5$ film was deposited by reactive direct-current magnetron sputtering (200 W) from a vanadium target (99.9%). The flow rates of oxygen (99.999%) and argon (99.9%) are 2 sccm and 19 sccm, respectively, with a volume ratio of 0.105. At the same time, they are introduced into the chamber, and the working pressure is maintained at 1 pa. The samples were annealed for 30 or 60 min at 750 °C in the same chamber. Finally, the samples were naturally cooled for two hours and then taken out of the vacuum chamber.

The crystallographic studies were carried out with the help of glancing incidence X-ray diffractometer (XRD, Bruker D8, Bruker, Karlsruhe, Germany) using a Cu K$\alpha$ radiation source ($\lambda$ = 1.5406 Å). The morphology of the films was investigated by scanning electron microscopy (SEM, Hitachi S-4800, Hitachi, Tokyo, Japan). A micro-Raman spectrometer (Renishaw RM2000, Renishaw, Gloucestershire, UK) was used with Ar$^+$ Laser (532 nm) as excitation source and a diffraction grating of 1800 gr.mm$^{-1}$ as monochromator to study the vibration modes of the synthesized sample [23]. A thermoelectrically cooled charge coupled device (CCD) camera was used as the detector for the Raman spectra in a back-scattering configuration. Reflection spectrum in reflection geometry were recorded by a UV–Visible (UV-Vis, 2401PC, Avantes, Apeldoorn, Netherlands) spectrometer in the range of 190–900 nm at room temperature.

## 3. Results and Discussions

The XRD patterns of three typical samples: A-0, A-30 and A-60, are shown in Figure 1a). Here, 0, 30 and 60 (with a unit of minutes) represent the annealing-time of the sample. It is hard to distinguish the diffraction peaks in the prepared samples (A-0), which indicates that nanograins are small. As the annealing time extends to 30 min, a weak diffraction peak appears at 32.44°. The peak deviates slightly form the $(\overline{1}22)$ peak at 32.05° of triclinic $AlVO_4$ in the standard card (JCPDS#39-0276). This is consistent with existing reports in the literature. Among the aluminum-vanadium oxides, it is known that triclinic $AlVO_4$ is the only stable ternary compound in the solid phase reaction of $V_2O_5$ and $Al_2O_3$ [24,25]. The compound has isostructural with $FeVO_4$. Extending the annealing time to 60 min, the strength of the $(\overline{1}22)$ peak enhances slightly. Using the Scherrer formula, the grain size of $AlVO_4$ is estimated as ~13.4 nm (A-30) and ~20.1 nm (A-60) from the diffraction peaks. The appearance of the new phase is also confirmed in the Raman spectrum, as illustrated in Figure 1b. A broad peak appears at 228–258 cm$^{-1}$ in Raman spectrum when the annealing-time is longer than 30 min. By the Voigt fitting [26], peak positions are

deduced as 246 cm$^{-1}$ and 245 cm$^{-1}$ for A-30 and A-60 samples, respectively, which is very close to the Raman peak value 242 cm$^{-1}$ of AlVO$_4$ phase reported by L. E. Briand et al. [27]. In Figure 1b, the Raman peak locating at ~300 cm$^{-1}$ is associated with Al$_2$O$_3$/Si and its strength decreases in A-series samples as the increase of annealing-time (or the interfacial mutual-diffusion).

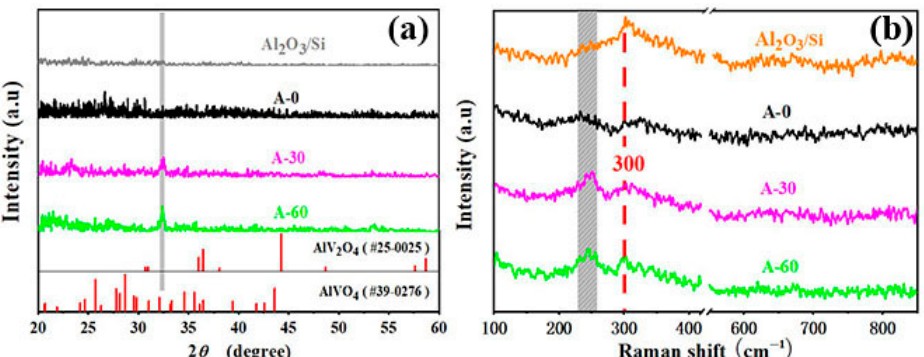

**Figure 1.** XRD patterns (**a**) and Raman spectrum; (**b**) of A-series samples with different annealing-time. Al$_2$O$_3$/Si annealed at 750 °C for 60 min is used as a reference. 0, 30 and 60 (with a unit of minutes) represent the annealing-time. The standard XRD data from JCPDS of AlV$_2$O$_4$ and AlVO$_4$ are attached for a comparison.

As shown in Figure 2a, the mutual-diffusion between V$_2$O$_5$ and Al$_2$O$_3$ during deposition results in a dense-particle morphology of an as-grown sample (A-0). Some bright-white points with an average diameter of ~110 nm were sparsely distributed on the surface of the sample. The bright-white points are the local spots with a large lattice distortion on the surface of an as-grown sample. As shown in an enlarged SEM image of Figure 2a (an area without bright-white points purposely selected), the dense-particle feature is clearly observed, from which the density of particles is estimated as ~185.3 particles per micrometer square. As presented in Figure 2b, under a 30 min post-annealing, the bright-white points are barely to see, and the density of particles is statistically ~185 particles per micrometer square. Extending the annealing time to 60 min, as displayed in Figure 2c, the bright-white points disappear completely, and the density of particles is ~185.4 particles per micrometer square. An SEM image of Al$_2$O$_3$/Si surface is also provided in Figure 2d as a reference.

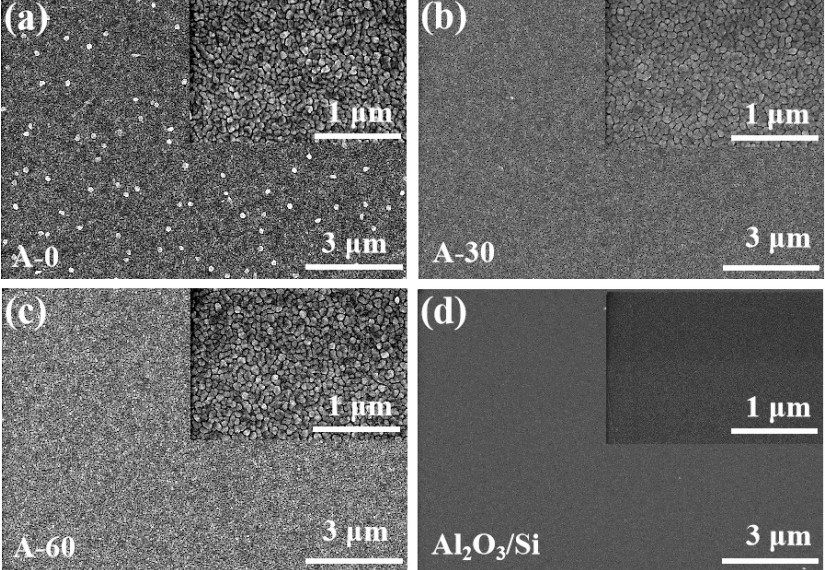

**Figure 2.** SEM images of A-series samples with different annealing-time. (**a**) The sample in preparation state (A-0). (**b**) The sample with 30 min post-annealing (A-30). (**c**) The sample with 60 min post-annealing

(A-60). (**d**) $Al_2O_3$/Si annealed at 750 °C for 60 min. The corresponding enlarged picture is inserted in each image.

It is speculated that the grains shown in the SEM images Figure 2a–c are not single particles or crystals; they are agglomerates of smaller sub-particles. The reaction that takes place during sputtering is a non-equilibrium reaction which results in heterogeneity in the products. In this respect, the structure is semi-dendritic, which is formed in non-equilibrium reactions in the presence of high electrical potential [28–30]. It was found that the short period post-annealing promotes the growth of crystallite size rather than the enlargement of the grain size. Although the dense-particle morphology of the sample is not significantly modified by a short-period post annealing, the disappearance of bright-white points indicates an existence of local-structure/local-phase modifications. The chemical heterogeneity decreases with thermal annealing while any micro-structure associated with the chemical heterogeneity also decays [28,29]. A short period annealing could prompt the new phase appearance by eliminating the local structure distortion, especially at the interface/surface. Referring to the results shown in Figure 1, it is considered that the appearance of $AlVO_4$ phase with high conductivity [24–27,31,32] is accompanied by the disappearance of bright-white points.

In Figure 3, the visible band reflectivity of the three typical A-series samples is presented. To guarantee the reliability of the test, we have used two silicon wafers to calibrate the light paths before the measurements. The difference of two reflective lights has been well suppressed in ±0.001 Abs. Next, a silicon wafer in one of the optical paths is removed to acquire the absolute reflectivity of the silicon wafer. The relative reflectivity of A-series samples is measured with a reference silicon wafer, as shown in Figure 3a). The reflectivity spectra of three typical A-series are plotted in Figure 3b. The highest and lowest reflectivity of Si wafers appear at 360 nm (42%) and 548 nm (22.5%), respectively, which is similar to that in literature [33,34]. When the wavelength is greater than 650 nm, the reflectivity of A-0 samples exceeds 60%, and the minimum reflectance (20%) appears at the wavelength of ~500 nm. In the wavelength range of 410–540 nm, the reflectivity of A-0 sample is lower than that of Si wafers. After annealing for 30 min, the reflectivity of A-30 sample is lower than the wavelength range of Si wafer and extends to 375–700 nm. The lowest reflectivity at 500 nm is close to 5%. However, when the annealing time increases further, the reflection spectra of A-60 and A-30 samples in the wavelength range of 375–700 nm are very similar. To evaluate the post-annealing effect on the reflectivity, a parameter $\eta$ is introduced as (R(A-0)-R(sample))/R(A-0) and is also plotted in Figure 3b). $\eta$ of A-30 (or A-60) increases abruptly in the range of 350–450 nm, reaches the highest value ~0.66 (470 nm, A-30), and then varies smoothly at long wavelengths, which means that the reflectivity suppression by the post-annealing is more pronounced at long wavelengths. In addition, $\eta$ gets its highest value in the wavelength range of 410–540 nm; that is, the lowest value of reflectivity is achieved as the relative change induced by post-annealing reaches the maximum. Finally, there is something else to be aware of that simply extending the annealing-time to increase the size of the nanograins does not necessarily result in better reflectivity suppression. As demonstrated in Figure 3b, the reflectivity spectrum of A-30 and A-60 are quite similar, but the former one has the lower value especially in the long wavelength range.

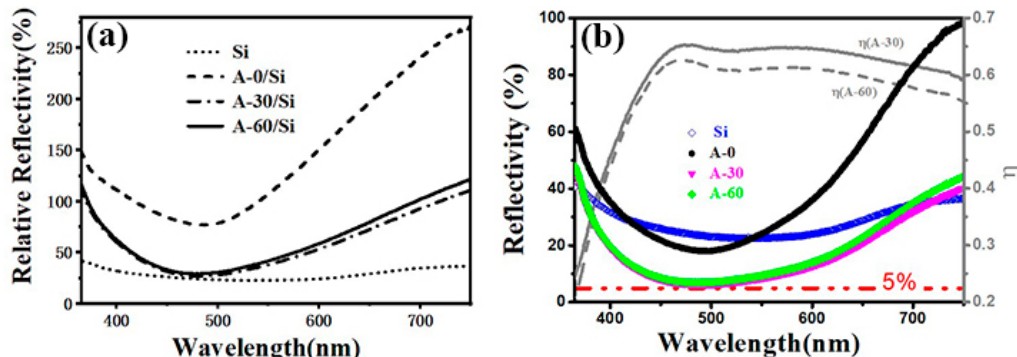

**Figure 3.** The reflectivity spectrum of A-series samples in the visible band. (**a**) Relative reflectivity spectrum of A-series samples. The relative reflectivity spectrum of the silicon wafer is also plotted together as a reference. (**b**) Reflectivity spectrum of A-series samples and silicon wafer. $\eta$ is defined as (R(A-0)-R(sample))/R(A-0) to illustrate the annealing effect on the variation of reflectivity.The red dotted line indicates the reflectivity of 5%.

## 4. Conclusions

Here, we have studied the effect of high temperature annealing on the reflectivity of $Al_2O_3/V_2O_5$ bilayer films in the visible light band. The main results are as follows: (1) $V_2O_5/Al_2O_3$ bilayer films were prepared by magnetron sputtering and annealed at high temperature to promote the mutual diffusion of the bilayers. Vanadium alumina nanocomposites containing $AlVO_4$ phase were successfully synthesized. (2) The surface morphology of the sample is a collection of dense-particles. This is a kind of non-equilibrium reaction which occurs during sputtering. Especially when vanadium is multivalent in this case, it will lead to the non-uniformity of the product, and this morphology is hardly affected by annealing. (3) The reflectivity of the sample was obtained by measuring the relative reflectivity of the sample and silicon. It was found that the wavelength range of the annealed sample with lower reflectivity than silicon wafers extended to 375–700 nm, and the lowest reflection depth was 5%. It is suggested that the significant suppression of the reflectivity of vanadium alumina nanocomposites thin film in visible band may originate from the appearance of $AlVO_4$ nanophase. In addition, reflectance suppression has stability in a certain annealing time. There are few relevant studies on high-temperature influences on the optical properties of ultrathin vanadium-alumina nanocomposites films. It was shown that the suppression of the reflectivity is significantly suppressed in visible band and hence such materials can be used in photocatalysis, gas-sensors and as absorption materials. The conclusion may be extended to lower frequencies. Our results are helpful regarding the applications of vanadium-alumina nanocomposites in photocatalysis and absorbents of electromagnetic radiation, including microwave frequencies for radar absorption [29,35] in catalytic plasma reactions [29].

**Author Contributions:** Conceptualization, Q.W. and H.W.; Methodology, H.W.; Validation, Q.W. and H.W.; Investigation, Q.W.; Data curation, Q.W.; Writing—original draft preparation, H.W. and Q.W.; Writing—review and editing, H.W. and Q.W.; Supervision, H.W.; Project Administration, G.Z. and H.W.; Funding Acquisition, G.Z. and H.W. All authors have read and agreed to the published version of the manuscript.

**Funding:** This work is supported by the National Key Project of Research and Development (No. 2021YFB3200102) and National Natural Science Foundation of China (62071312).

**Institutional Review Board Statement:** No applicable.

**Informed Consent Statement:** No applicable.

**Data Availability Statement:** The data presented in this study are available on request from the corresponding author Hai Wang.

**Conflicts of Interest:** The authors declare no conflict of interest.

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
