# Peer review of "Ultra-Low Reflectivity in Visible Band of Vanadium Alumina Nanocomposites"

_coatings, doi:10.3390/coatings12091276_

Round 1

Reviewer 1 Report

1.Kindly restructure the abstract as like this flow aim, materials and methods, results and conclusion.
2.Kindly check once all the keywords should be in running text.
3. if there any human samples tested in the experimentation.
4. need more explanation in the sample preparation.
5. P value is less than 0.05.
6. Research gap is not clear.
7. Include the following reference to enhance the manuscript.

A. Parthiban, V. Vijayan, S. Dinesh Kumar, L. Ponraj Sankar, N. Parthipan, Dawit Tafesse and Mebratu Tufa ‘
Parameters of Porosity and Compressive Strength-Based Optimization on Reinforced Aluminium from the Recycled Waste Automobile Frames, Advances in Materials Science and Engineering, Volume 2021 |Article ID 3648480 | https://doi.org/10.1155/2021/3648480, pp. 1-10. 

S. Jayaprakash, S. Siva Chandran, Bhiksha Gugulothu, R. Ramesh,  M. Sudhakar and Ram Subbiah
Effect of Tool Profile Influence in Dissimilar Friction Stir Welding of Aluminium Alloys (AA5083 and AA7068), Advances in Materials Science and Engineering, Volume 2021, Article ID 7387296, https://doi.org/10.1155/2021/7387296. pp 1-12.

Reviewer 2 Report

The topic of synthesis and characterization for new materials is very important and the paper can be accepted after the following corrections and suggestions have been carefully met:

1-    Please revise the language carefully.

2-    You have to state the main result in the abstract.

3-    Revise the keywords.

4-    The introduction has to be extended with new references for different materials, for example you can start as:

“Synthesis and characterization of new materials is very important topic for several applications [Refs], some new references such as:

-       "A significant role of MoO3 on the optical, thermal, and radiation shielding characteristics of B2O3–P2O5–Li2O glasses." Optical and Quantum Electronics 54, no. 2 (2022): 1-19.

-       "Optical and radiation shielding effectiveness of a newly fabricated WO3 doped TeO2–B2O3 glass system." Radiation Physics and Chemistry (2022): 109968.

-       "Synthesis, physical and nuclear shielding properties of novel Pb–Al alloys." Progress in Nuclear Energy 142 (2021): 103992.

-       "ZnO–Bi2O3 nanopowders: Fabrication, structural, optical, and radiation shielding properties." Ceramics International 48, no. 3 (2022): 3464-3472.

- "CdSe supported SnO2 nanocomposite with strongly hydrophilic surface for enhanced overall water splitting." Fuel 321 (2022): 124086.

-  "Stainless steel supported NiS/CeS nanocomposite for significantly enhanced oxygen evolution reaction in alkaline media." Journal of Solid State Electrochemistry (2022): 1-12.

-       "Synthesis and characterization of B2O3-Ag3PO4-ZnO-Na2O glasses for optical and radiation shielding applications." Optik 248 (2021): 168199.

5-    Revise the order of the equations, figures, and tables throughout the whole manuscript.

6-    The explanation of some figures is very poor.

7-    State all the important results in the conclusion.

Reviewer 3 Report

Hello, 

it is impressive to see Vanadium alumina nanocomposites are reducing reflection. can authors comment on the composition of V2O5: Al2O3 the annealing compound.

Reviewer 4 Report

Review of the Manuscript ID: coatings-1815040

Qiujin Wang, Guozhong Zhao, and Hai Wang; Ultra-low reflectivity in visible band of vanadium alumina nanocomposites

This manuscript reports a composite which achieves ultra-low reflectivity in the visible band.  Although the paper is interesting and it should be published, it has several important omissions and corrections before it can be published.  As part of this review, I cite several references (R1  to R6) at the end of this text for the completeness of the paper.

1.    Introduction

1.1.        In the introduction, the reason for the use of alumina as support for V2O5 should be explained. That is:  V2O5 suffers from poor conductivity and stability for many applications and hence several studies investigated supported V2O5 to overcome these shortcomings [R1]. The improvements in V2O5 performance as regards mechanical and  thermal stability are also observed through supports such as Al2O3, SiO2, TiO2 [R2].

2.    Experiments

2.1.        Experimental section is too brief and hence causes a lot of speculation as regards the process. In particular:

(a)  Line 60-61:  The working pressure of oxygen 60 (99.999%) and argon gas was 1.0 Pa with a mass flow ratio 10.5%  -  

This statement needs clarification.  Is it the mass ratio of oxygen and argon or infact is there a gas flow constantly fed into the sputtering chamber?   If this is the case, what are the flow rates and what kind of mass flow controllers were used to keep the flow rates constant and pressure at 1.0 Pa?

(b)  Irrespective of what exactly the experimental conditions are, the mass ratio cannot be 10.5%; it has to be stated in terms of volume (or molar) or mass ratio at 10.5 (without %).

(c)   Sample preparation for SEM:  As stated above, V2O5 has low conductivity and hence it may not be possible to use SEM at high magnifications.   The question is:  Are the samples coated with carbon or Au/Pd? 

 3. Results and Discussion

3.1. Line: 114 - 117

From the above observations, it is found that the short period post-annealing promotes the growth of grain size rather than the enlargement of the particle size. Although the dense-particle morphology of the sample does not be significantly modified by a short period post annealing; however, the disappearance of bright-white points indicates an existence of local-structure/local-phase modifications.  ……

The term “Grain” refers to the primary units of the agglomerated particles or the relatively large (several micrometre in size) domains, usually in metals,  which can be observed comfortably by SEM at the magnifications used by the authors or even by optical microscopy.  However, what is measured   by SEM at magnification ca. 10k is ca. 185 particles per micrometer square from which the average size of the sputtered particles can be calculated approximately by assuming they are all spherical and closely packed.  Such a calculation shows that the grain size is about 80 nm which is very similar to that obtained directly from the SEM images.

However, SEM images indicate that the particles are themselves composed of smaller particles.  Hence these grains are agglomerated particles.  But, the nature of these sub-particles cannot be evaluated because the magnification is not sufficiently high.  I assume that higher SEM magnifications have not been used because the conductivity of the samples was not sufficient and they were not coated. 

What is measured  by XRS using Scherrer equation  is the crystallite size not grain size.  They are  ca. 13 nm and ca. 20 nm for A-30 and A-60 samples,  illustrating the effect of annealing as expected.     

3.2. The reaction that takes place during sputtering is a non-equilibrium reaction which results in heterogeneity in the products especially when the metal, vanadium in this case, has multi-valency.  The implications of non-equilibrium reactions as regards the product heterogeneity has recently been examined [R3 and R4] and a mechanism proposed [R4, R5] which clearly applies to this paper as indicated below:

(a)  Grain structure:  The grains shown in the SEM images Figs. 2 (a,b,c) are not single particles or crystals, they are agglomerates of smaller sub-particles. In this respect, the structure is semi-dendritic.  These structures form during non-equilibrium reactions in the presence of high electrical potential as shown in the references R3 and especially R4. Mechanism of structure formation is available in R4 and R5.

(b) Bright spots:  I believe these structures are the result of chemical heterogeneity and the bright spots  have lower electrical conductivity. This can of course be resolved by performing EDX-spectroscopy on the samples during SEM measurements.  The size of the bright spots is more than sufficient for EDX-spectroscopic measurements. Absence of EDX-spectroscopy is infact a major detriment.

(c)The chemical heterogeneity decreases with thermal annealing while any micro-structure associated with and the chemical heterogeneity also decays as shown in the references R3 and R4.  The same conclusion is apparent in Fig. 2.  

4. Conclusions

It is shown that the  suppression of the reflectivity is significantly supressed in visible band and hence such materials can be used in photocatalysis, gas-sensors and as absorption materials.  No mention of the frequency range of absorption is made. The conclusion can be extended to lower frequencies.  It is known  [References R4 and R6] that such materials would be good absorbents of electromagnetic radiation including microwave frequencies for radar absorption [R4, R6] as well as in catalytic plasma reactions  [R4].

References cited by the referee:

R1.  Briand, L.E.; Tkachenko, O.P.; Guraya, M.; Gao, X.; Wachs, I.E.; Grünert, W. Surface-Analytical Studies of Supported Vanadium Oxide Monolayer Catalysts. J. Phys. Chem. B 2004, 108, 4823–4830.

R2. Banares, M.A.; Martínez-Huerta, M.; Gao, X.; Wachs, I.E.; Fierro, J.L.G. Identification and roles of the different active sites in supported vanadia catalysts by in situ techniques. In Studies in Surface Science and Catalysis; Corma, A., Melo, F.V., Mendioroz, S., Fierro, J.L.G., Eds.; Elsevier: Amsterdam, The Netherlands, 2000; Volume 130, pp. 3125–3130.

R3.  Akay G. Co-Assembled Supported Catalysts: Synthesis of Nano-Structured Supported Catalysts with Hierarchic Pores through Combined Flow and Radiation Induced Co-Assembled Nano-Reactors. Catalysts. 2016; 6(6):80.

R4. Akay, G. Plasma Generating—Chemical Looping Catalyst Synthesis by Microwave Plasma Shock for Nitrogen Fixation from Air and Hydrogen Production from Water for Agriculture and Energy Technologies in Global Warming Prevention. Catalysts 202010, 152.

R5. X Chen, CJ Hogan Jr, Nanoparticle dynamics in the spatial afterglows of nonthermal plasma synthesis reactors, Chemical Engineering Journal 2021, 411, 128383.

R6. M. Green and X. Chen, Recent progress in nanomaterials for microwave absorption. J. Materiomics, 5, 2019, 503-541.   

Round 2

Reviewer 4 Report

The review by the authors has been done  very well.  The revised manuscript can be published as is.

Author Response

Thank you very much.